# Study the Electrical Properties of Surface Mount Device Integrated Silver Coated Vectran Yarn

**DOI:** 10.3390/ma15010272

**Published:** 2021-12-30

**Authors:** Abdella Ahmmed Simegnaw, Benny Malengier, Melkie Getnet Tadesse, Gideon Rotich, Lieva Van Langenhove

**Affiliations:** 1Department of Materials, Textiles and Chemical Engineering, Faculty of Engineering and Architecture, Ghent University, B-9052 Ghent, Belgium; Benny.Malengier@UGent.be (B.M.); Lieva.VanLangenhove@UGent.be (L.V.L.); 2Ethiopian Institute of Textile and Fashion Technology, Bahir Dar University, Bahir Dar 1037, Ethiopia; melkie.getnet@bdu.edu.et; 3Industrial and Textile, School of Engineering and Technology, South Eastern Kenya University, Kitui 90215, Kenya; rotichgideon2016@gmail.com

**Keywords:** conductive yarn, silver-coated Vectran, surface mount device, E-yarn, electrical resistance, smart textile, wearable electronics

## Abstract

Smart textiles have attracted huge attention due to their potential applications for ease of life. Recently, smart textiles have been produced by means of incorporation of electronic components onto/into conductive metallic yarns. The development, characterizations, and electro-mechanical testing of surface mounted electronic device (SMD) integrated E-yarns is still limited. There is a vulnerability to short circuits as non-filament conductive yarns have protruding fibers. It is important to determine the best construction method and study the factors that influence the textile properties of the base yarn. This paper investigated the effects of different external factors, namely, strain, solder pad size, temperature, abrasion, and washing on the electrical resistance of SMD integrated silver-coated Vectran (SCV) yarn. For this, a Vectran E-yarn was fabricated by integrating the SMD resistor into a SCV yarn by applying a vapor phase reflow soldering method. The results showed that the conductive gauge length, strain, overlap solder pad size, temperature, abrasion, and washing had a significant effect on the electrical resistance property of the SCV E-yarn. In addition, based on the experiment, the E-yarn made from SCV conductive thread and 68 Ω SMD resistor had the maximum electrical resistance and power of 72.16 Ω and 0.29 W per 0.31 m length. Therefore, the structure of this E-yarn is also expected to bring great benefits to manufacturing wearable conductive tracks and sensors.

## 1. Introduction

Smart textiles are textiles which contain advanced technologies enabling them to sense and give a response or react to the conditions of the environment [1]. They are textiles enabling different functions and features that can improve textile materials performance and the wearer experience [2]. Most smart textiles are manufactured by using the integration of electronic components or conductive fibers onto textile substrates [3]. The development of electronic and photonic components for integration in fabrics will increase the application potential of electronics textiles [4]. Electronic textile or E-textile is one of the under-development technologies with a wide potential impact on wearable, flexible, conformable, lightweight, and electronic devices integrated on textile clothing [5]. They can also be used in large-area electronic systems embedded in technical textile [6,7], for the healthcare sector [8,9], for on-body communication [10], and for entertainment and fashion, sport activity tracking, space suits, and security monitoring applications [11,12].

Most electronic textiles available today are made by attaching either permanent or removable electronic functionality [13,14,15]. The combination of microelectronics into textile substrate can be carried out integrating to a yarn by various means [16], e.g., through weaving [17], through sewing and embroidering [18,19], using hybrid soldering techniques [20], using sewing integration [21], applying electrically conductive inks [22], through 2D-screen printing [23], and through 3D-printing [24].

Mechanical connectors such as snap buttons, crimp connections, crimp flat-pack, socket buttons, bolt connection, ribbon cable connectors, and hook and loop have been used for connecting electronics devices onto the surface of the textile fabric [14]. However, those forms of interconnection still suffer from incompatibilities including soft, flexible, pliable textile fabrics and rigid electronic components. This also has a significant impact on the final design and properties of the textile fabric. Therefore, the integration of electronics devices could be done without compromising the final design and characteristics of the textile substrate.

Surface mount device (SMD) components are now so small that they can be embedded in yarns with hardly a change in diameter. Goros and Weg [25] developed a highly elastic tubular ring protective sleeve from small portable electronic devices. In addition, Horvath et al. [26] fabricated a pressure sensor barometer on a custom double-sided flexible and stretchable printed circuit board by using SMD and casting the assembly in urethane rubber to obtain a sensing sleeve for use as a cardiac compression device. Hardy et al. [27,28,29] presented integration methods of seven-strand copper electronic yarn embedded with semiconductor dies or micro electromechanical systems (MEMS).

The electrical characteristics of conductive yarns are an important design factor that is influenced by the physical, electro-mechanical, and chemical properties of the material [30,31]. Electrically conductive textile material manufactured from a conductive yarn requires a perfect characterization for the design and manufacturing of safe and consistently reliable products.

Different researchers have studied the effects of physical and environmental factors on the electrical conductivity of conductive yarn. For instance, the influence of washing on the electrical resistance of conductive yarns [32] and the electrical resistance of textile transmission lines after pre-treatment processes have been recently discussed. Mola [33], Liu et al. [34], and Sahin et al. [35] have studied the effect of laundry on the electrical conductivity of stitched surface mount e-textiles and fabrics for e-textile applications. The analytical results found that the electrical conductivity of the base fabrics significantly affected the mechanical stress of laundry conditions. In addition, textile wet chemical processes caused a significant impact on the electrical resistance of conductive textile materials that were used as conductive track and transmission lines for the applications of e-textile.

A study by Bogan et al. [36] discussed the influence of abrasion on electrical conductivity of blended enameled copper alloy filaments. The results confirmed that the electrical resistance value started to increase slowly and then tended to reach a point at which it began to increase quickly with failure soon occurring. Ding et al. [37] examined the effect of temperature on the conductivity of knitted fabrics embedded with silver-coated conductive yarns and detected that the electrical resistance decreased with increasing temperature due to the physical contact of the overlapped plied silver-coated conductive yarns, which caused a decrease in contact resistance.

However, as mentioned above, the majority of these E-yarns were manufactured by using enameled copper conductive strands and these typically have highly rigid and coarse metallic yarn interconnections, which were the direct reason for their low durability in repeated deformation and unacceptable harsh handling of the resultant fabrics for clothing. Moreover, to the best of the authors’ knowledge, there are no studies that examine both the effect of the physical-mechanical (gauge length, extension, and abrasion) and environmental factors (thermal, moisture, and chemical) on the electrical conductivity of both conductive and E-yarns.

The purpose of this research work was to integrate SMD resistors into SCV conductive yarn and perform an analysis with respect to electro-mechanical properties. The optimal design and manufacturing of such types of E-yarn were investigated in different types of specimen gauge lengths and incorporation of microelectronics. The developed E-yarn brought the property of being easy to process for manufacturing of smart textile conductive tracks and sensors via knitting, weaving, braiding, and embroidery without difficulty. These E-yarns can be used in a wide range of applications such as manufacturing of conducting track, communication, and sensors. Hereafter, the experimental set up is given and the effects of yarn gauge length, strain, abrasion, temperature, solder pad length, and washing on the electrical resistance of the resulting E-yarn were investigated.

## 2. Materials and Methods

### 2.1. Materials

In this study, 15 Tex multifilament SCV (Liberator^®^ 40) conductive yarn purchased from Syscom (1305 Kinnear Rd., Columbus, OH, USA) was used for the experiment [38]. According to the supplier information, the coated yarn was manufactured by coating the high-performance liquid-crystal polymer (polyester-polyacrylate fiber) with two thin layers of silver, created by Kuraray Ltd., Japan, and marketed as silver-coated Vectran, Liberator. The properties of the Vectran conductive thread are shown in Table 1, and its cross-section is illustrated in Figure 1A. In addition, a Cermet resistor 68 Ω with 1% tolerance, 0.27 mm length with 0.04 mm solderable metallic terminal pads (Samsung Electro-Mechanics RC2012F1820CS SMD 0805 resistor) was chosen as the electrical component to integrate with the yarn (Figure 1B). As solder paste, a carbon conductive paste was used (i.e., 847 solder series part number 473-1230-ND) from MG chemicals.

### 2.2. Methods

The methodological approach for this research comprised two phase. Initially, the development of SMD resistance embedded electronic yarn was performed. Thereafter, the effects of factors on the electrical resistance of the E-yarn were studied based on the given experimental setup according to the international standard guidelines.

#### Development Process of Conductive Yarn with Embedded SMD Resistor

The integration of the SMD resistor into SCV conductive yarn was performed using a Beta V3 PRO reflow temperature-controlled programmable machine (manufactured from Bay 98, Shannon Free Zone, Shannon, Co. Clare, Ireland). For connecting the SMD to the conductive yarn, a vapor phase reflow soldering method was used. This is a non-contact heating mechanism and constant heat circulation inside the area of application can be easily performed. The creation of a robust and efficient bond between the solder pads of the SMD resistor and SCV conductive yarn to form an interconnection is one of the key steps of the integration process.

In order to hold the SMD resistor and conductive yarn tightly during the soldering process, a miniature template was prepared with a wooden cupboard as shown Figure 2. This wooden cupboard was manufactured using a fully automated high-speed and high-precision laser cutting and engraving machine. The wooden cupboard was chosen due its thermal resistance, non-solderability, and ease of construction through laser cutting. The design of the cupboard was carried out based on the end-use requirement of the SMD resistor embedded E-yarn: a circuit model, which is configured by a series of resistors.

The three main steps followed during the construction were:

Conductive yarn placement: The first stage of the reflow process was the placement of two pieces of SCV conductive yarn, 20 cm in length, and the SMD resistor onto the wood template by manual pick-and-placing using a pair of tweezers based on a simple manual x-y plane sliding concept. The position of the sample in the wooden cupboard is shown in Figure 2.

Solder paste printing: The second step was covering the cupboard template with a precise alignment pre-designed perforated commercial metallic stencil, which has different holes based on the SMD size. Then, depositing and printing of 100-μg carbon conductive lead-free solder paste was performed on the metallic terminal part of the SMD resistor and the tip of conductive yarn using a syringe pad-by-pad mechanism. Thereafter, the wooden sample holder was placed into the inside chamber of reflow solder oven.

Reflow soldering: After the solder paste printing, the actual vapor phase of the reflow soldering process was performed and manufacturing of the prototype was carried out. This was achieved using a benchtop reflow oven as shown in Figure 3A. The reflow oven follows a specific temperature profile as shown in Figure 3B which is compatible with the manufacturer’s recommendations of carbon-conductive assembly soldering paste (red line), the RC2012F1820CS SMD resistor (blue line), as well as the SCV conductive yarn. At 160 °C, the thermal energy from the heat source started to melt the solder paste and reflow began. Afterwards, an intermetallic compound formed due to the reaction between solder paste, metallic terminal of the SMD resistor, and the outer metallic layer of the SCV conductive yarn. Finally, strong solder joints were formed on the connection between the terminal points of the SMD resistor pads and the conductive yarn, as shown in Figure 4.

### 2.3. Experimental Set Up

#### 2.3.1. Measurements of Length Dependent Electrical Resistance

The characterization of the electrical properties of the conductive yarn samples was carried out by using standardized methods for conductivity measurements in accordance with the AATCC TM84 test method by the combination of a digital multimeter and Burster clamping four-point resistance measuring as shown in Figure 5. The current applied to at the ends of the conductive yarn and the output voltage was measured. The length-dependent electrical resistance of the sample was evaluated at variable gauge lengths (i.e., 0.05, 0.1, 0.15, 0.2, 0.25, 0.3, 0.35, 0.4, 0.45, and 0.5 m) of conductive yarn. Ten samples were measured and the results were analyzed by Analysis of Variance (ANOVA) methods.

#### 2.3.2. Measurements of Strain Dependent Electrical Resistance

The stress-dependent electrical resistance of the SMD integrated SCV E-yarn under tension was obtained by modifying a conventional Instron 3369 universal strength-tester machine. The bottom and upper jaws were clamped at both ends of the E-yarn. This experiment used a guideline standard ISO 2062 [39] with a pretension of 2 N force, a gauge length of 200 mm, and a strain rate of 5 mm/min. At the SCV E-yarn tips, the two electrodes of the digital multimeter (resolution of 1 µV) were attached and measured the resistance (R) of the conductive yarn as a function of current (I) input and voltage (V) output set up as described in Equation (1) with 0.25% extension performed.

Furthermore, the resistance of the SCV E-yarn without externally applied load was measured after strain was released and with a relaxation time of 5 min. Finally, to obtain the sensitivity of the yarn, the resistance change ratio (RCR) was calculated using Equation (2). The yarn that has high gauge factors will have high sensitivity and it can be used in the application of strain sensors. Five samples were measured to obtain average results. The electrical resistance under 0.25% strain difference was measured and recorded. To find a trend, the results were analyzed by using ANOVA.
(1)$$R=\frac{V}{I}\;$$
(2)$$\mathrm{RCR}=\frac{R_{f}-R_{0}}{R_{0}}\;$$
where R_0_ is the initial resistance (the resistance before stretching) and R_f_ is the final resistance (the resistance after released from stretching). A positive value of the RCR means the resistance after removing the load is larger than the initial resistance, while a negative RCR means the resistance after removing the load is smaller than the initial resistance. Both positive and negative RCR are hysteresis effects.

#### 2.3.3. Measurements of Abrasion

The conductivity behavior of the SCV E-yarn was examined after being subjected to mechanical abrasion in order to investigate their reliability and durability. Eight yarn specimens running under 0.5 N tension at test speeds of 10 mm/min were tested simultaneously with a Shirley yarn abrasion tester. When the E-yam broke, the flexible holder fell and a signal was sent to the control unit recording the number of revolutions required to break the yarn. After the E-yarn was subjected to abrasion, the electrical resistance of SCV E-yarn was examined with a 0.001-mV-resolution digital multimeter. Twenty measurements were taken and the effects of abrasion on resistance of SCV conductive yarn and E-yarn was analyzed by paired *t*-test techniques.

#### 2.3.4. Measurements of Temperature Dependent Electrical Resistance

The SCV conductive yarn and E-yarn were heated by oven at 20 °C interval ranges from 0 °C to 100 °C for 10 min. After heating took place, the samples were left to cool down for 120 s under standard laboratory conditions. The corresponding changes in electrical resistance with changing temperature level were measured by a digital multimeter and the results were recorded. The linear resistance of conductive yarn at a given temperature can be expressed as Equation (3).
(3)$$R_{T}={\;R}_{0\;}[1+\alpha_{R\;}\left(T-T_{0\;}\right)]{\;}_{\;}$$
where $R_{T}$ is the total resistance at temperature T, $R_{0}$ is the resistance at $T_{0}$. For this experiment, $T_{0}$ was set at room temperature (20 °C). In addition, $\alpha_{R}$ stands for temperature coefficient resistivity of the conductor. To determine whether the electrical resistance is constant or not under change of temperature, ANOVA was applied.

#### 2.3.5. Effect of Washing

The conductive yarns and E-yarns underwent a washing procedure in order to examine their long-term reliability behavior against repeated washing. According to the international standard AATCC 2014 [40], the samples were washed 25 times at 20 °C for 8.5 min washing per cycle and 5 min drying cycle with a spin speed of 500 rpm. The selected procedure was typical home laundering with a HWM 140-9188S Haier automatic washing machine filled with 15 L of water and fully dissolved 100 g of detergent. The automatic washing was performed at machine agitation speed of 119 rpm with consecutive action of delicate washing, rinse, and drying with a final spin speed of 500 rpm for 5 min. In addition to the conductive yarn and E-yarn samples, the machine was filled with cotton polyester blended fabrics to reach its standard load of 2 kg. After each washing cycle, the samples were dried at room temperature for 12 h. Finally, three samples of each cycle were taken and the electrical resistance was measured with the four-point-probe. To find a trend and determine the effects of laundering on the resistance of both SCV conductive yarn and E-yarn, ANOVA analysis was implemented for evaluating the results.

#### 2.3.6. Measurements of Total Electrical Resistance of E-Yarn

The electrical conduction mechanism sequence of SCV E-yarn illustrated as an equivalent resistor chain model is shown in Figure 6. A series electrical circuit was used to measure the low magnitude of resistance. The E-yarns were placed in series configuration at the Burster clamping four-point probe devices as shown in Figure 6. The total resistance of the SMD implanted E-yarn was determined using Kirchhoff’s 2nd law, where a constant current across the E-yarn was applied and the voltage drop down at each specific node in the circuit was measured by using an internal resistance 0.001 accuracy volt meter. With this series electrical circuit, the resistance of each node was calculated as in Equation (1) and the total resistance of the E-yarn calculated as in Equation (4). Five repeats of each sample were measured and the average value was presented.
(4)$$T_{R}={\;R}_{\mathrm{ly}}+R_{\mathrm{Ic}}+{\;R}_{\mathrm{SMD}}+R_{\mathrm{rc}}+{\;R}_{\mathrm{ry}}\;$$
where${\;T}_{R}$ is the total electrical resistance of the E-yarn,${\;R}_{\mathrm{ly}}$ is the left conductive yarn resistance,${\;R}_{\mathrm{lc}}$ is the left connector resistance, $R_{\mathrm{SMD}}$(which was 68 Ω) is the SMD resistor resistance,${\;R}_{\mathrm{rc}}$ is the right connector resistance and${\;R}_{\mathrm{ry}}$, is the right conductive yarn resistance.

#### 2.3.7. Measurements of Electrical Power

In addition to electrical resistance, power is a phenomenon in wearable electronics. Since the SCV conductive yarn acted as a resistance at the circuit, shown in Figure 6, the power dissipated at the conductive threads and also at the SMD resistor are important factors in joules heating (also called ohm heating or resistance heating). According to Joule-Lenz’s law, the heat output P (W) produced by the conductor material is proportional to the product of the square of its resistance R (Ω) and the direct current I (A). The first phase of the experiment was verified by applying a variable voltage into the two ends of silver conductive Vectran thread and the E-yarn by using a four-point probe clamping device with an EL301R power supply source. The voltage drop between the two ends was measured by a 0.001-accuracy multimeter. Afterwards, the resistance was calculated by using Equation (1) based on Kirchhoff’s second law. Five samples were examined with a variable input voltage (U) starting from 0 volts to their registered failure of current flow in the circuit and the results were presented.
(5)$$P=\mathrm{VI}\;\mathrm{or}\;P={\mathrm{RI}}^{2}\;$$
where P, is power, R, is resistance, I, is current and V, is voltage drop.

#### 2.3.8. Effects of Solder Pad Overlap Thickness on Resistance and Its Resistance Ratio

During the soldering process, there was an overlap of the terminal part of the SMD resistor and the SCV conductive yarn. An intermetallic bond or solder pad was formed due to the reaction between the solder paste, the terminal of the SMD resistor, and the conductive yarn, as shown in Figure 7. For utilizing the conductive connections between the conductive track lines and SMD elements, it was important to examine and optimize the resistance of the solder pad.

Where c is the length of the conductive yarn sample, b is the joint overlap length of SMD tips and conductive yarn, X is the total length of the E-yarn.

The joint overlap length b (soldered pad specimen length) and the electrical resistance of each solder pad was measured by using digital calipers and a four-point probe at room temperature, respectively. The electrical resistance of each soldered pad was compared with the electrical resistance of an equal length of the conductive yarn and results were calculated in a resistance ratio. The resistance ratio was calculated for each joint by Equation (6).
(6)$$R_{r}=\frac{R_{C}}{R_{\mathrm{CY}}}\;$$
where $R_{r}$ is the resistance ratio, $R_{c}$ is the resistance of the solder pad connector, and $R_{\mathrm{CY}}$ is the resistance of same length in conductive yarn per a given mm length. The resistance of the connecter was calculated by Equation (7).
(7)$$R_{b}=\frac{1}{2}\left(R_{\mathrm{Ey}}-2R_{a}-R_{R}\;\right)\;$$
where$\;R_{\mathrm{Ey}}$ is the total resistance of E-yarn over the total length x (see Figure 7, so including the joints), $R_{R}$ is the resistance of the resistor, and $R_{a}$ is the resistance of the conductive yarn.

A resistance ratio of 1 indicates that the resistance of solder pad joint is equivalent to an equal length of a solid conductive yarn. A resistance ratio of less than 1 shows that the solder joint has a lower electrical resistance than an equal length of conductive yarn; similarly, a resistance ratio greater than 1 indicates a higher electrical resistance in the soldered joint than in an equal length of conductive yarn. Finally, the joint overlap thickness with five different cross sections were measured and the effects of overlap thickness on the electrical resistances of the E-yarn were explored and assessed by using ANOVA.

## 3. Results and Discussion

### 3.1. Electrical Conductivity

The dependence of yarn resistance (R) on yarn gauge length for individual samples are shown in Figure 8. It proves that the electrical resistance increased with increasing clamping gauge length as claimed in Ref. [41]. A nearly linear trend is visible, indicating the SCV conductive yarn has an appropriate uniform makeup with nearly constant resistance per unit length of 2.802 Ω/m.

The graph in Figure 8 proves that as the test gauge length increased, the resistance of the yarn increased proportionally. The variable factor was the gauge length level. From Table 2, it can be seen that the dependent of electrical resistance on gauge length approached a linear function in the form f(x) = 2.802 x + 0.0204. The *p*-value was for the slope < 0.001. Hence, the result shows that the gauge length on the sample influences the electrical resistance (statistically significant), and the *p*-value of the intercept of 0.003 was significant. In this case, the correlation coefficient of the curve R^2^ = 0.994 and Probe > F = 0.000 described that the electrical resistance of the yarn is greatly correlated with length.

### 3.2. Electrical Resistance under Strain and after Strain

The electrical resistance of the SCV E-yarn after cyclic straining is displayed in Figure 9, which shows RCR at different levels of strain after stretch and relaxation of the SCV yarn. The fractional increment in resistance $\Delta R/R_{0}$ varied almost linearly with the applied strain (positive gauge factor).

Furthermore, to investigate the mean resistance value after straining, a single-factor ANOVA was performed. It can be reported that the electrical resistance was influenced by cyclic straining, which was highly significant at α = 0.05 level-value for the slope and intercept (*p*-value < 0.001) (statistically significant). It also described that the electrical resistance of the yarn is greatly correlated with cyclic strain ($R^{2}$ = 0.99). The reason of increasing resistance after stretching is due to the elastic deformation that leads to an increase in length and increase in density of a conductive yarn whereby current flow will be hindered and overall resistance increases [42].

When the SCV conductive yarn is stretched, the tiny, coated silver particles on the surface of the conductive yarn become far apart and hence the resistance of the conductive yarn will be increased [43]. For wearable e-textile applications, the conductive yarn will not able to stretched to such high strains. They are rather used for repeated stretching at lower strains.

### 3.3. Effects of Abrasion

The effects of mechanical abrasion on the electrical conductivity of both SCV conductive yarn and E-yarns are illustrated in Figure 10.

Figure 10 shows the dependency of electrical resistance on abrasion. It was determined by applying a paired ANOVA test with α = 0.05. The statistical analysis showed that abrasion influences the electrical resistance of SCV conductive yarn and SCVE-yarn. The difference in electrical resistance before and after the yarn exposed to abrasion was highly significant (*p* < 0.001) as shown in Table 3. This is due to the formation of scratches and cracks in the outer metallic layer of the SCV conductive yarn. In addition, the mechanical stress during the abrasion may cause deformation of the conductive yarn surface, perhaps causing a combined effect of creating projecting fibers (formation of protruding fiber) and breakage at the SCV filament fiber strands. This was proven by using optical microscope as shown in Figure 11. The significant change in electrical resistance started at 225-abrasion cycle for SCV and at 150-abrasion cycle for E-yarn, which can clearly withstand the abrasion. The electrical resistance of the conductive yarn slowly increased at a magnitude of only 0.71% before 225-abrasion cycle. However, the resistance of the E-yarn was increased by 61.5% at 225-abrasion cycle. In addition, after the 800-abrasion cycle, the effects of mechanical abrasion were increased by a magnitude of 114.6% and 240.9% of the SCV conductive yarn and E-yarn respectively. The gradual increase in electrical resistance values shows that electrical resistance values often increase, meaning that the conductivity of the conductive yarns generally decreases after mechanical wear. This might be because of the mechanical tensions that occurred during abrasion procedures. Furthermore, the SCV E-yarn started to deteriorate faster than original SCV yarn due to extra damage that occurred at the solder pads. Therefore, the E-yarn could be covered by using a protective, insulated material such as encapsulation with *t**hermoplastic polyurethane* (TPU) or silicon encapsulation. These will protect the surface of the E-yarn from external harsh environments and the durability and performance will also increase.

### 3.4. Effects of Temperature

The effect of temperature variation on the electrical resistance of the SCV yarn and the E-yarn were investigated and the outcomes are shown in Figure 12. The change of electrical resistance with temperature was presented and detected at two different phases i.e., the temperatures below 50 °C, and temperatures between 50 °C to 100 °C. In the first phase, at the beginning of the heating, the resistance of the conductive yarn was approximately constant, with the curve showing only a slight increase up to 50 °C. After passing this critical temperature, the electrical resistance increased more quickly relative to beforehand.

The electrical resistance of the SCV conductive yarn and E-yarn was abruptly increased between the temperature range of 50 °C to 100 °C with a magnitude of 47.49% and 57.99% respectively. This is because when thermal energy is given to the SCV yarn, the resulting vibrations of the particles hinder the electron movement. Each collision uses up some energy from the free electron and this is the basic cause of resistance change [44]. In addition to these, the coefficient of thermal expansion of the conductive yarn increased with temperature increased. This electrical behavior at elevated temperatures may perform as a wearable fire-detecting sensor for fire protection.

Based on the graph in Figure 12, the SMD integrated SCV E-yarn had greater resistance change relative to its input material. This is not only due to the collision of electrons in the SCV conductive yarn, but also due to an increase in the junction temperature at the SMD, which restricted the current further. Therefore, the resistance of the SMD embedded E-yarn changed. The statistical analysis at α = 0.05 showed that the dependence of electrical resistance on temperature was statistically significant (*p* < 0.001) for both the SCV yarn and E-yarn. The electrical resistances of these two samples rose with increasing temperature.

### 3.5. Effects of Washing

The effects of washing on electrical resistance percentage of SCV conductive yarn and SCV E-yarn are shown in Figure 13.

From Figure 13, it can be seen that starting from 0 to 10 washing cycle, the variation tendency of the washing cycle on electrical resistance of both the SCV conductive yarn and E-yarn was small. There was a small increase in electrical resistance for both samples after each washing cycle and their resistance increased by a magnitude of 6% for SCV conductive yarn and 10% for E-yarn before 10 wash cycles. The statistical result at alpha = 0.05 showed that electrical resistance increased slightly with each washing cycle and it was statically significant (*p* < 0.001). After 15 washing cycles, at alpha = 0.05, the statistical analysis confirmed that the dependent resistance on the washing cycle determined by ANOVA was highly significant (i.e., *p* < 0.001). In addition, the regression analysis showed that the effects of washing on electrical resistance gradually grew after 10 washing cycles. The R^2^ = 0.99 showed that there was a strong positive correlation between the washing cycle and electrical resistance. This was due to the effects of laundering on the individual fibers, which caused movement and damage across the SCV-conductive structures. Part of the mechanical action during washing is similar to abrasion presented before, and which also increases the resistance. Furthermore, washing might result in surface fractures on the SCV-conductive yarns as well as deformation along the SMD transmission lines. As a result of this, there would be an increase in electrical resistance or conductor line discontinuities. After each washing cycle, the electrical resistance of all samples was increased [45]. After 25 wash cycles, the electrical resistance of both the SCV-conductive yarn and the E yarn increased by 77.4% and 91.94%, respectively.

### 3.6. Total Electrical Resistance of E-Yarn

The total electrical resistance of the E-yarn is a direct function of the resistance of left SCV-conductive yarn, left connector solder pad resistance, the SMD resistor, the right connector solder pad, and the right SCV-conductive yarn. Therefore, the average total resistance of E-yarn was measured based on Equation (4) and the result is depicted in Table 4. The practical experiment results showed that the total electrical resistance of SCV E-yarn was 72.16 Ω per 0.31 m length.

### 3.7. Power of Conductive SCV Yarn and E-Yarn

In addition to the electrical resistance characterization, performance of the conductive threads and E-yarn was examined in terms of power and change of electrical resistance with current levels [46]. Figure 14 presents the performance of the SCV threads and E-yarn for their mean maximum attainable power before failing. It was evident that at variable input voltage source between 0 V and 5 V and the power of the SCV-conductive thread and E-yarn increased continuously. From Figure 5 it can be observed that both the SCV-conductive yarn and E-yarn were able to tolerate higher powers of 1.18 W and 0.29 W respectively. These indicated that, at a lower voltage source, the conductive thread delivered 1.18 Jules of energy per second. The E-yarn performed well up to 4.5 V without deterioration and then failed. Since many wearable sensors perform on continuous voltages as low as 0.8 volts (stepped-down as efficiently as possible from a 3-volt source) [47], these manufactured E-yarns can be proposed for the development and application of wearable electronic textiles.

### 3.8. Effect of Solder Pad Overlap Thickness on Total Electrical Resistance of E-Yarn

The total electrical resistance of the E- yarn is a direct function of the resistance of left SCV-conductive yarn, left connector solder-pad resistance, and resistance of the SMD resistor, right connector solder pad, and right SCV-conductive yarn. Therefore, the average total resistance of E-yarn is 72.16 Ω per 0.31 m length. The resistance ratio R_R_ with one mm solder pad (connector) overlap length is 66.58%. The effect of solder pad overlap size on the total resistance of the E-yarn and the resistance ratio are shown in Figure 15A,B, respectively.

Figure 15A,B show that the resistance of E-yarn slightly increased when the solder joint pad overlap length increased from 0.5 to 5 mm. It is hypothesized that there will be an addition of solder paste and overlap on the solder pad length, which increases the connector thickness and the resistance. In addition, there is the probability of the formation of an amorphous region in the micro pad connector. The existing results showed that the resistance of a solder pad was in the range of 1.588 to 2.125 Ω and was long-time stable on mechanical abrasion. When the connector overlap thickness increased from 1 mm to 5 mm, the total resistance of the E-yarn increased and the resistance ratio increased from 65% to 72%. The practical result showed that the resistance ratio of the solder pad overlap thickness with 0.5 to 0.5 mm was still less than 1 (i.e., the resistance of solder pad is less than the resistance of conductive yarn). Therefore, the electrical conductivity of the solder pad joint is functional.

Table 5 summarizes the dependence of electrical resistance of SCV E-yarn on solder-pad length using ANOVA. The prob. > F = 0.000 shows that the overlap size of the solder pad was statistically significant on the electrical resistance of SCV E-yarn. Furthermore, the R^2^ = 0.984 indicates that there was a positive correlation between solder pad overlap thickness and the electrical resistance [48]. From visual observation, it was shown that if the solder pad thickness was much smaller than the pad size, the electrical connection usually failed due to the loss in mechanical bond connection.

## 4. Conclusions

In this research work, the method used for the integration of SMD into the SCV conductive yarn was vapor phase soldering methods for the application of smart textiles. The integration of SMD resistors into the SCV conductive yarn was performed without damaging them. This integration method replaced other mechanical connectors such as snap buttons, crimp connections, crimp flat-pack, socket buttons, bolt connection, ribbon cable connectors, hook and loop, and conductive adhesives. The effect of yarn gauge length, cyclic strain, abrasion, temperature, washing, and solder-pad length on the electrical resistance of SCV-conductive yarn and SMD resistor-embedded SCV E-yarn based on international testing standards were analyzed.

The application relevance and the advantage of resisting physical and external environmental factors was the reason for this research investigation. The integration of the SMD resistor into textile-conductive yarn helps to develop for the manufacturing of sensors, actuators, conductive tracks, and the fully functional wearable electronic textile.

It is evident that as the testing gauge length increased, the electrical resistance of conductive yarn increased. The resistance of conductive yarn also increased with strain due to the elastic deformation SCV-conductive yarn, which led to an increase in length and increase in density of the conductive yarn. The analytical findings showed that at α = 0.05, level, the effects of strain on the resistance of SCV conductive yarn and E-yarn were highly significant.

The effects of external factors such as mechanical abrasion, temperature, and washing environments on the electrical conductivity of E-yarn were also presented and the analytical outcomes indicated that the electrical resistance of both SCV-conductive yarn and E-yarn increased significantly with the increase in abrasion cycle, temperature, and washing cycle. Furthermore, the experimental results showed that the manufactured E-yarns were functional after 800 mechanical abrasion cycles and 25 washing cycles.

In order to escape these challenges and to increase the durability and reliability of the SCV E-yarn, the overall structure could covered within TPU film or using silicon encapsulation or by bridging by nonconductive adhesives and the coated silver conductive particle could firmly adhere to the yarn surface. All these protections should be efficient against mechanical stresses during mechanical abrasion, temperature, laundering, and mechanical strain, which had the biggest impact on the conductivity. In addition, the solder pad overlap thickness was directly influenced the electrical resistance of the SCV E-yarn. The change on the gauge factors and its sensitivity on the electrical resistance of the SCV E-yarn due to the external factors indicated that the constructed SCV E-yarn can be used to manufacture different applications of wearable sensors. However, these effects must be taken into account.

In the future, it will be vital to look into the effect of physical elements such as the conductive yarn’s structure, count, density, fineness and twist, as well as effect of the geometry of the conductive yarn on the electrical conductivity of the E-yarn.

## Figures and Tables

**Figure 1 materials-15-00272-f001:**
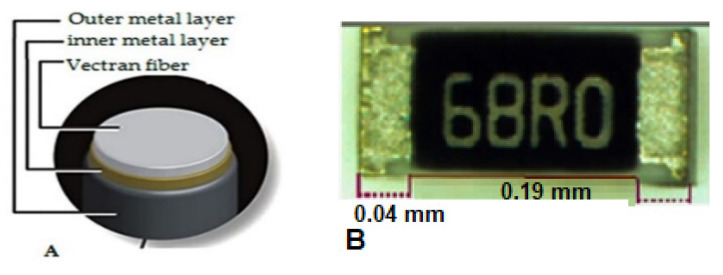
Silver coated Vectran (Liberator^®^ 40) (**A**), 68 Ω Cermet resistor SMD (**B**).

**Figure 2 materials-15-00272-f002:**
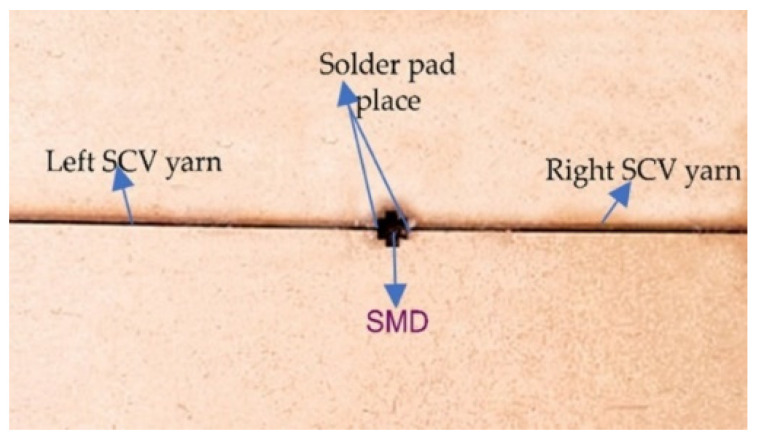
Cupboard design in series configuration.

**Figure 3 materials-15-00272-f003:**
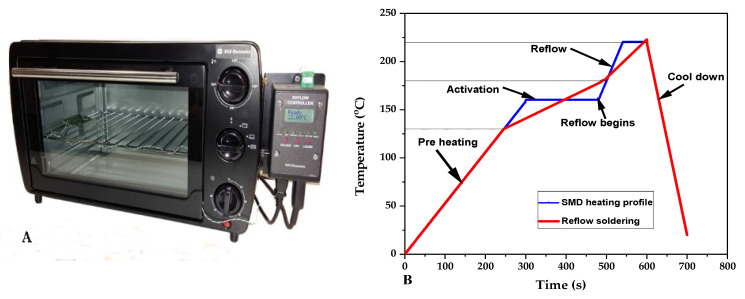
Bench top of the reflow oven (**A**), temperature profile of the reflow oven (**B**).

**Figure 4 materials-15-00272-f004:**
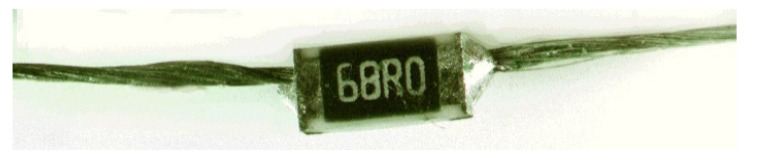
SMD integrated E-yarn.

**Figure 5 materials-15-00272-f005:**
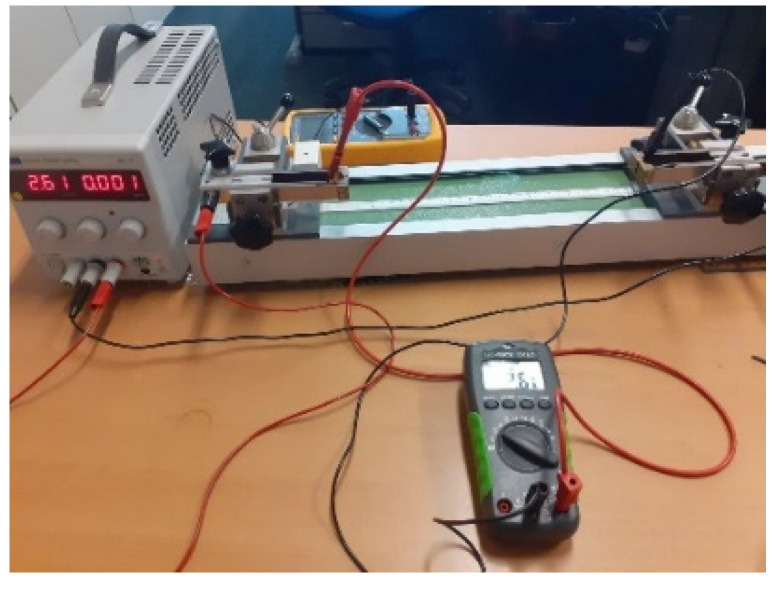
Measuring of length-dependent resistance (four-point clamping device).

**Figure 6 materials-15-00272-f006:**
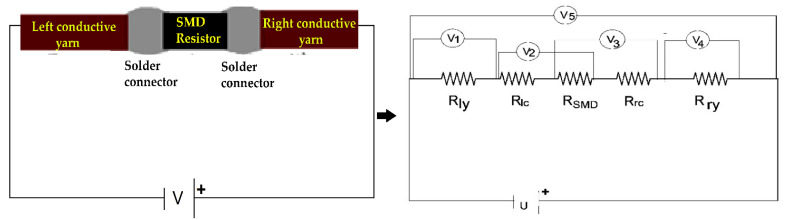
Schematic diagram of different resistors found in the E yarn and measurements of voltage drop down at each node.

**Figure 7 materials-15-00272-f007:**
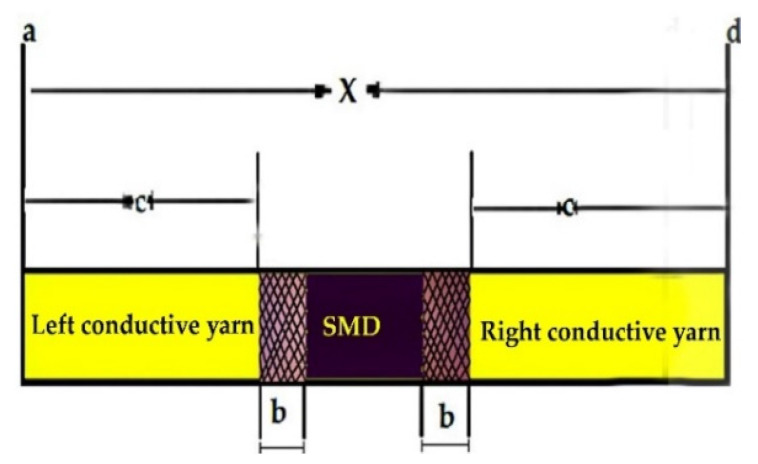
Sketch illustration of calculation of resistance ratio.

**Figure 8 materials-15-00272-f008:**
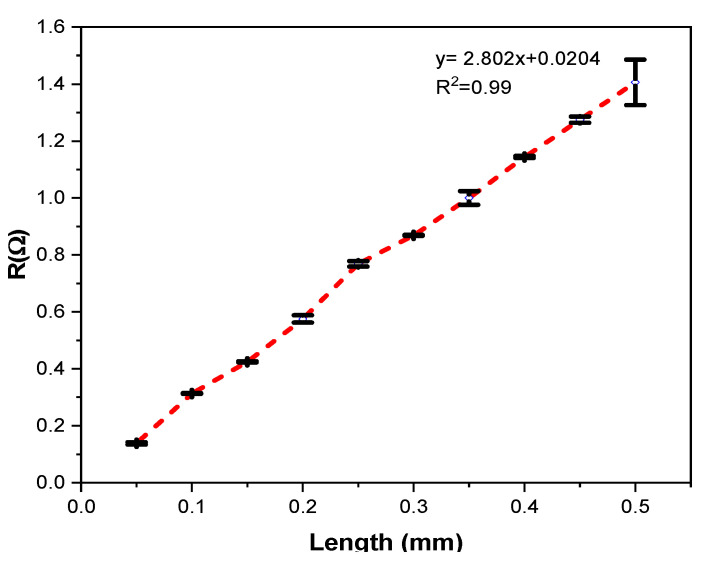
Effects of length on resistance of SCV conductive yarn.

**Figure 9 materials-15-00272-f009:**
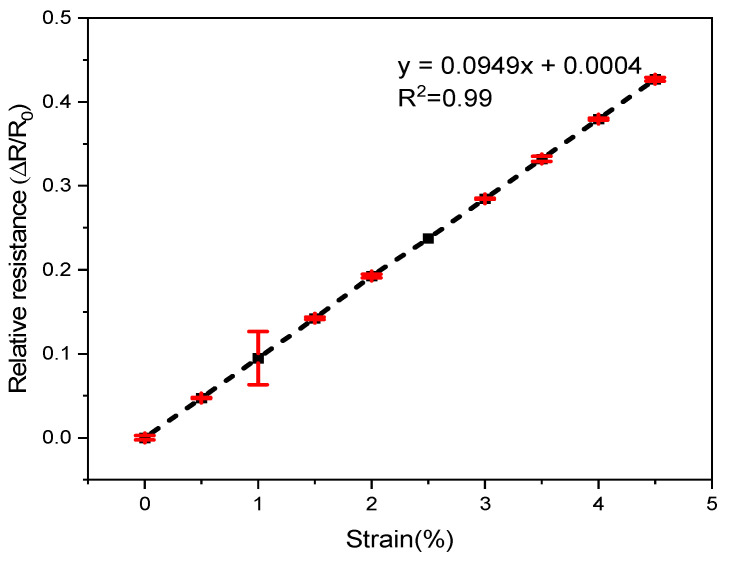
Effects of strain on electrical resistance.

**Figure 10 materials-15-00272-f010:**
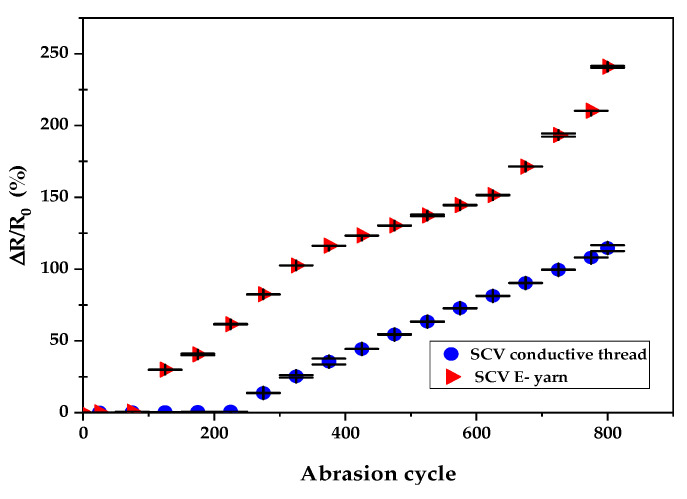
Effects of abrasion on electrical resistance.

**Figure 11 materials-15-00272-f011:**
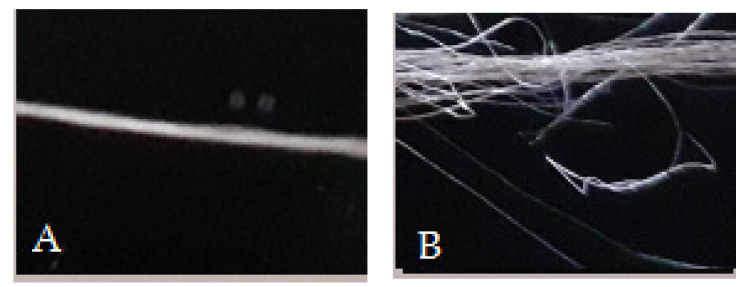
SCV conductive yarn before (**A**) and after (**B**) abrasion.

**Figure 12 materials-15-00272-f012:**
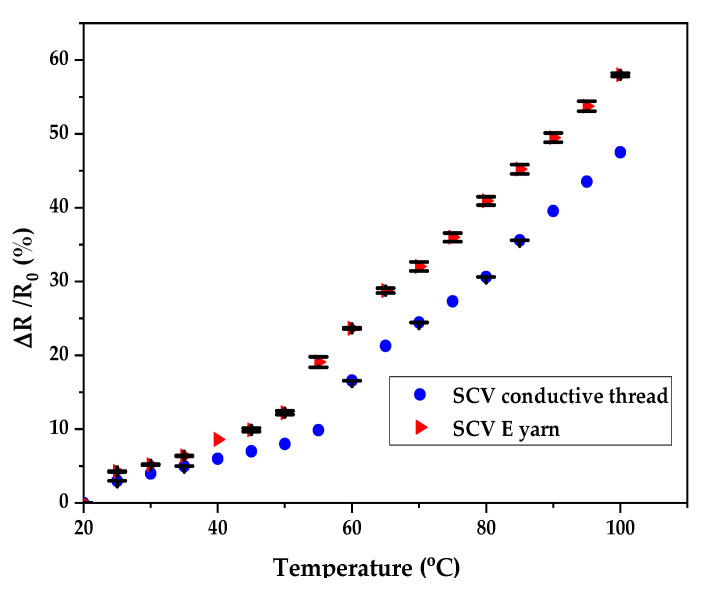
Effects of temperature in electrical resistance.

**Figure 13 materials-15-00272-f013:**
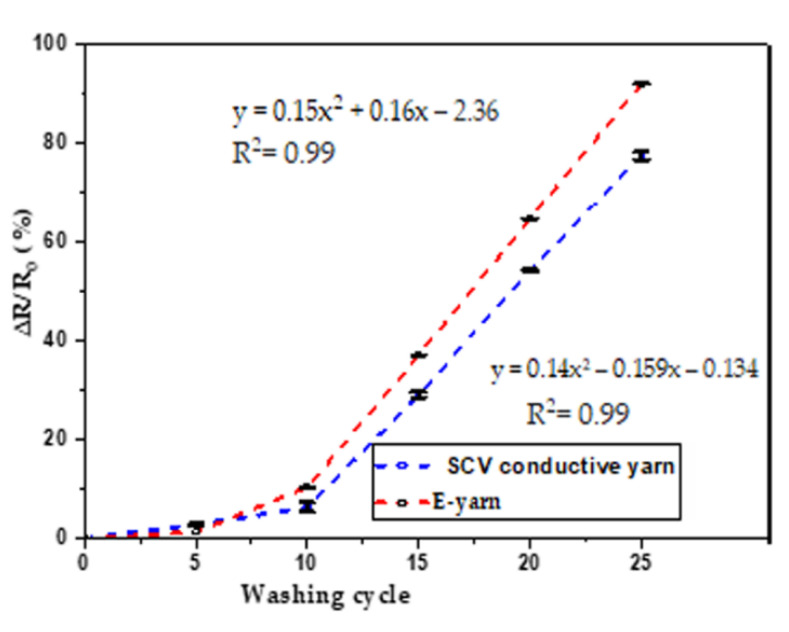
Effects of washing on resistance of conductive yarn and E-yarn.

**Figure 14 materials-15-00272-f014:**
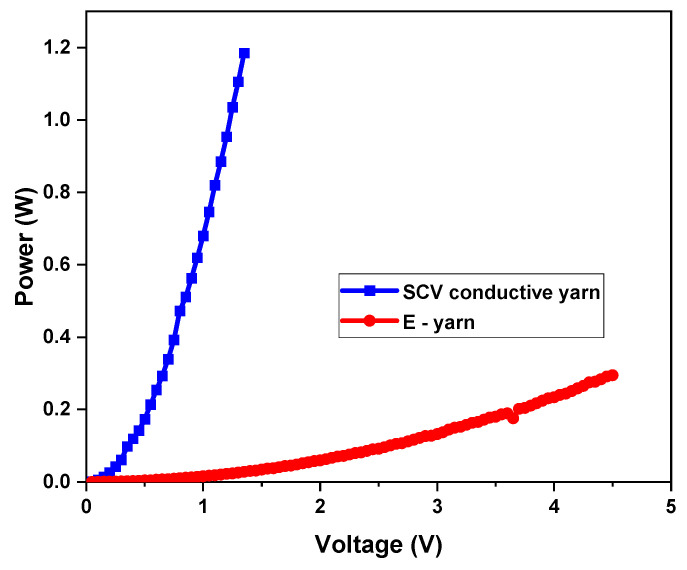
Power depicted in SCV conductive yarn and E-yarn.

**Figure 15 materials-15-00272-f015:**
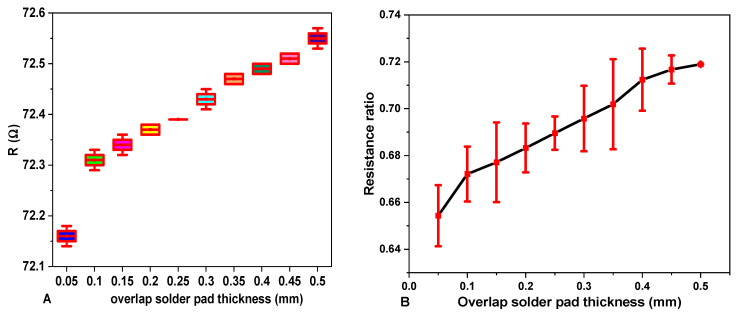
Effects of solder pad connector overlap thickness on total resistance of E-yarn (**A**) and its resistance ratio (**B**).

**Table 1 materials-15-00272-t001:** Overview of Vectran (Liberator^®^ 40) conductive yarn.

Composition	Brand Name	Structure	Tex	Yarn Diameter (mm)	Metal by Weight (%)	Operating Temperature (°C)	Melting Point (°C)
Silver-coated Vectran	Liberator^®^ 40	multifilament	15	0.226	82.1	Up to 200	350

**Table 2 materials-15-00272-t002:** Regression overview of length-dependent electrical resistance of SCV conductive yarn.

Source	SS	df	MS	Number of obs = 100		
Model	16.17	1	16.17	Prob > F = 0.00		
Residual	0.10	98	0.001	R-squared = 0.99		
Total	16.27	99	0.164	Root MSE = 0.03.		
**Resistance**	**Coef.**	**Std. Err.**	**t**	**P > t**	**[95% Conf. Interval]**	
Length	2.8000	0.023	124.23	0.000	2.750	2.850
_cons	0.0204	0.007	3.04	0.003	0.007	0.035

**Table 3 materials-15-00272-t003:** Statistical analysis of effects of abrasion on electrical resistance.

	Source	SS	DF	MS	F	Prob. > F	R^2^	t	P > |t|
SCV	Between groups	105,208.52	26	4046	3461.54	0.00	0.99	129.96	0.00
	Within groups	63.14	54	1.16					
	Total	105,271.6	80	1315.89					
E-yarn	Between groups	214,198.2	26	8238.4	13,969.4	0.000	0.95	42.0	0.000
	Within groups	31.8	54	0.589					
	Total	214,230	80	2677.88					

**Table 4 materials-15-00272-t004:** Total electrical resistance of E-yarn.

Sample	Left Conductive Yarn		Right Conductive Yarn		E-Yarn		
	Voltage Drop (V)	R(Ω)	Voltage Drop (V)	R(Ω)	Voltage Drop (V)	R(Ω)	Connector R(Ω)
1	0.0680	0.425	0.049	0.306	0.720	72.00	1.634
2	0.0680	0.425	0.048	0.300	0.719	71.90	1.588
3	0.0670	0.419	0.049	0.306	0.719	71.90	1.588
4	0.0701	0.438	0.050	0.313	0.730	73.00	2.125
5	0.0680	0.425	0.048	0.300	0.720	72.00	1.638
Av.	0.0682	0.426	0.049	0.305	0.722	72.16	1.71

**Table 5 materials-15-00272-t005:** Overview of the statistical of effects of solder-pad thickness on conductivity.

Source	SS	DF	MS	F	Prob. > F	R^2^	t	P > |t|
Between groups	1.352	19	0.711	2388.79	0.000	0.984	48.88	0.000
Within groups	0	20	0					
Total	1.352	39	0.035					

## Data Availability

The data presented in this study are available on request from the corresponding author.

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
