# Peer review of "Study the Electrical Properties of Surface Mount Device Integrated Silver Coated Vectran Yarn"

_materials, 2021, doi:10.3390/ma15010272_

Round 1
Reviewer 1 Report
In article entitled “Study on the effects of physical and environmental factors on 2 the electrical resistance of SMD integrated Silver coated Vec-3 tran yarn” authors did comprehensive analysis of integration of SMD resistor with silver coated Vectran (SCV) yarn. Authors perfumed studies of electrical resistance dependence on length and strain, as well as abrasion effects, temperature and washing. All experiments are perfumed on statistical suitable set of data with appropriate discussion. However, in humble opinion of this reviewer, presented study is more appropriate for 2 or 3 conference papers, rather than one journal article. I cannot see scientific contribution, what was main hypothesis that has been answered? For example, after all tests, authors concluded in lines 403-404 with “Therefore, in order to achieve higher reliability the solder pad overlap thickness and size have to be optimum.” Is there any guidance, based on performed study, to achieve that goal? Moreover, is there any result that was not expected, lines 415-419, and is there any scientific explanation for that?
Electrical characterization of conductive threads should be expanded in terms of power and change of electrical resistance with current levels. Please take a look on these papers:
https://doi.org/10.3390/ma14123390
https://doi.org/10.3390/electronics10080967
Authors should be encouraged to resubmit with emphasized scientific contributions.
Author Response
Thank you for giving us the opportunity to submit a revised draft of the manuscript “Study on the effects of physical and environmental factors on the electrical resistance of surface mount device integrated Silver coated Vectran yarn” for publication in Materials. We appreciate the time and effort that you and the reviewers dedicated to providing feedback on our manuscript and are grateful for the insightful comments on and valuable improvements to our paper. We have incorporated most of the suggestions made by the reviewers. Those changes are highlighted within the manuscript. Please see below, in red, for a point-by-point response to the reviewers’ comments and concerns

Reviewer 2 Report
Authors in this work perform different tests (cyclic strain, abrasion, temperature, washing, etc.) in soldered SMD into conducting yarn. Although a lot of tests have been performed based on international standards the paper doesn’t add any significant information regarding material science. It would be interesting to the readers to see the dependency of yarn properties (dimensions, density, structure, etc.) in all these tests and add additional figures related to the other parameters as Figure 8. Authors also should further elaborate on their methodology regarding the calculation of the electrical resistance of the E-yarn and add some details in Figure 6. Finally, and although it is mentioned that the proposed method replaces other mechanical connectors a detailed literature review is missing indicating the benefits of the proposed methodology.
On paper logistics:
A lot of abbreviations are missing (e.g. SS, dF MS, e.t.c.).
Reference 36 is incomplete.
Numbering in equation (5) is missing.
Author Response

(The authors gave the same response as above.)

Reviewer 3 Report
The authors investigated the effects of physical and environmental factors (including strain, solder pad size, temperature, abrasion and washing) on the electrical resistance of SMD integrated Silver coated Vectran yarn. The structure of this E-yarn is also expected to bring great benefits to manufacturing wearable conductive tracks and sensors. Although being interesting, I find that there are some major issues with the paper that require addressing prior to this being considered for publication in this journal. I have identified the main points for consideration below:
- This manuscript has some spelling typos, style errors and grammatical errors. Pleases carefully check and correct them in the revised manuscript.
- The Full name of SMD should be given in the tittle.
- In the introduction section, some recent related references such as Nanomaterials 2021, 11, 1962 and Analytica Chimica Acta 1170 (2021) 338480 are recommended to be also cited.
- Error bars should be added in Figs. 8, 9, 13 and 14.
- The scientific explanations are not sufficient in the present manuscript. So, more scientific discussion is required in the revised manuscript.
Author Response

(The authors gave the same response as above.)

Round 2
Reviewer 1 Report
Authors improved manuscript in the proper way and I recommend publication.
Author Response
First of all, we appreciate for your valuable comments given and thank you for taking your precious time for the comments again. Your previous comments help us to improve our article. Thank you for your recomendations for our article to be publish in MDPI Material.
Reviewer 2 Report
The authors' work is more like a review paper, so they should include a lot of other works in the field from the literature and perform trade-off analysis and comparisons, as was indicated in the first review. Also, the authors didn't include to the reviewer's comments regarding the dependency of yarn properties (...and other comments) but they postponed for future work.
Author Response
First of all, we appreciate for your valuable comments given and thank you for taking your precious time for the comments again. Your previous comments help us to improve our article.
we thank for the comments given. This paper is an article paper, which discussed about the effects of physical and environmental factors on the electrical resistivity of electronic yarn and its input conductive yarn. We performed many experiment works.
The dependency of yarn properties (i.e. the density, structure, twist, count, evenness, irregularity, and other yarn parameter affect the conductivity. However, due to time constraint we cannot incorporate it at this moment. The conductive yarn that we used in this research only had a specific physical characteristic. We are not a manufacturer for the conductive yarn. Since the conductive yarn was manufacturing from Syscom advanced materials (http://www.metalcladfibers.com/), at this moment we cannot order to manufacture a conductive yarn with different physical characteristics. For future work, we will investigate the effects of the physical yarn property on the electrical conductivity.
We hope that the reviewer will understand our concern and will accept to publish in MDPI Material.

Reviewer 3 Report
Accept in present form.
Author Response
First of all, we appreciate for your valuable comments given and thank you for taking your precious time for the comments again. Your previous comments help us to improve our article. Thank you for your recommendation to be publish our article in MDPI Material.